# Study on the Pasting Properties of Indica and Japonica Waxy Rice

**DOI:** 10.3390/foods11081132

**Published:** 2022-04-14

**Authors:** Sicong Fang, Cheng Chen, Yuan Yao, John Nsor-Atindana, Fei Liu, Maoshen Chen, Fang Zhong

**Affiliations:** 1State Key Laboratory of Food Science and Technology, Jiangnan University, Wuxi 214122, China; 6190112022@stu.jiangnan.edu.cn (S.F.); feiliu@jiangnan.edu.cn (F.L.); fzhong@jiangnan.edu.cn (F.Z.); 2Science Center for Future Foods, Jiangnan University, Wuxi 214122, China; 3School of Food Science and Technology, Jiangnan University, Wuxi 214122, China; 4International Joint Laboratory on Food Safety, Jiangnan University, Wuxi 214122, China; 5Zhejiang Wufangzhai Industrail Co., Ltd., Jiaxing 310031, China; chencheng@wufangzhai.com (C.C.); yaoyuan@wufangzhai.com (Y.Y.); 6Department of Nutrition and Dietetics, School of Allied Health Sciences, University of Health and Allied Sciences, PMB 31, Ho 00233, Ghana; jansor@uhas.edu.gh

**Keywords:** indica waxy rice, japonica waxy rice, pasting property, protein

## Abstract

In this study, the physicochemical properties of indica (IWR) and japonica (JWR) waxy rice were investigated to find the critical factor that differentiates the pasting behaviors among the two cultivars. The results showed that the peak viscosity of 5 IWR flours was in the range of 1242 to 1371 cP, which was significantly higher than 4 JWR flours (667 to 904 cP). Correlation analysis indicated that all pasting parameters were not correlated (*p* < 0.05) with physicochemical properties of rice flours and the fine structure of isolated starches. The pasting profiles of IWRs were still significantly higher than those of JWRs after removing lipid, while there were no significant differences between the two cultivars after removing protein sequentially. Meanwhile, the addition of extracted protein from JWR to the isolated starch significantly decreased the viscosity compared to the addition of protein extracted from IWR. The protein composition results found that the IWR protein contained about 18% globulin and 64% glutelin, while the JWR protein contained 11% globulin and 73% glutelin. The addition of glutelin to isolated starch significantly decreased viscosity compared to the addition of globulin. Therefore, the differences in the content of globulin and glutelin might be the main reasons that differentiate the pasting behaviors of the two cultivars.

## 1. Introduction

Rice is one of the most popular staple foods in East Asian countries [1]. In general, rice is divided into waxy and non-waxy rice. Waxy rice is further divided into two types, namely indica and japonica waxy rice (IWR and JWR). The amount of amylose content present in rice is the main basis for classifying rice into these two broad categories: non-waxy and waxy rice. Non-waxy rice generally contains amylose content between 10 and 30% [2], while that of waxy rice is mostly below 5% [3]. Although the interest in rice research has gained considerable attention, nearly all the previous studies have mainly focused on the processing and cooking characteristics of non-waxy rice (indica and japonica rice) [4]; or comparing the physical and chemical properties and thermal characteristics of waxy rice with non-waxy rice [5]. Literature survey revealed that little attention has been paid to studying the differences between the two cultivars of waxy rice.

Previous studies demonstrated that pasting properties, which depend on extrinsic and intrinsic factors, affect cooking quality and, eventually, the final food product. The key intrinsic factors refer to the amylose content [6,7] and amylopectin fine structure [8] of starch, since starch accounts for about 80% of waxy rice. Amylose is a linear polymer, which has more crystalline regions and requires more energy to gelatinize. Therefore, amylose content was positively correlated to pasting temperature [2,9]. Due to the increased crystalline order, longer branch chains in amylopectin usually need more thermal energy to break the kinetic barrier [10]. Besides intrinsic factors, protein is an important extrinsic factor that affects pasting properties [11,12]. Xie et al. [13] showed that the gelatinization viscosity of waxy rice starch increased after proteolysis. However, Li et al. [14] and Martin et al. [15] found that the viscosity curves of waxy rice flour and non-waxy rice flour decreased significantly after protease or dithiothreitol treatment. These conflicting results showed that the effects of protein on the gelatinization of waxy rice need to be further investigated. Therefore, there is a need to study the pasting properties of the two cultivars of waxy rice, and then investigate the parameters that cause the properties to differ between the two cultivars of waxy rice.

In this study, we aimed to find the critical factor that causes the pasting properties to differ between indica and japonica waxy rice. The physicochemical properties of flours and the structural information of isolated starches of 5 IWRs and 4 JWRs were investigated. Meanwhile, the effects of lipid, protein and its composition on the pasting properties were also studied. The findings of the present study can be exploited by waxy rice-related manufacturers.

## 2. Materials and Methods

### 2.1. Materials

Two varieties of IWR, namely IWR-1 and -2, were obtained from Saraburi and Bangkok in Thailand, and the other three, namely IWR-3, -4, and -5, were collected from Hubei, Anhui, and Jiangxi in China, respectively. Four varieties of JWR, namely JWR-1, -2, -3, and -4 were collected from Anhui, Jiangsu, and Heilongjiang in China, respectively. Isoamylase (Cas: 9067-73-6, ∼200 units/mL) was obtained from Megazyme (Wicklow, Ireland). The alpha-amylase (0.15 Units/g) from Aspergillus oryzae was purchased from Sigma-Aldrich (St. Louis, MO, USA). All chemicals and reagents were of analytical grade.

### 2.2. Preparation of Waxy Rice Flour

Waxy rice flours were prepared using a dry grinding method [16]. In brief, 1.0 kg of rice grains were milled by a high-speed crusher (SS-1022, Shengshun, Jinhua, China). The flour (IWR/JWR-number-F) was passed through a 60-mesh sieve and stored in a glass bottle before further analysis.

### 2.3. Pasting Properties

The pasting properties of flour and starch samples were tested using a rotational rheometer (Discovery DHR-2; TA Instruments, New Castle, DE, USA) according to Park et al. [17] with slight modifications. A 40-mm diameter parallel-plate geometry was used and the gap geometry was set to be 1000 μm. Each sample (0.18 g, dry basis) was added to 1.3 mL deionized water to prepare rice flour dispersion or starch dispersion at a mass ratio of 12% (dry basis). After mixing for 15 min, the dispersions were all poured onto the Peltier plate of the rheometer. After loading the dispersions, a thin layer of silicone oil (Sinopharm, Shanghai, China) was added to the perimeter to prevent moisture loss during heating. Samples were firstly held at 50 °C for 1 min, and secondly heated at a rate of 12 °C/min to 95 °C and maintained for 5 min, and then cooled to 50 °C at a rate of 12 °C/min according to Zhu et al. [18], with some modifications. The pasting viscosity of the samples were recorded. All measurements were done in triplicate.

The temperature procedure for measuring the pasting properties of waxy rice flour by Rapid visco-Analyzer (RVA 4500, Melbourne, Australia) refers to the American Association of Cereal Chemists 61-02 method (AACC 2000). Rice flour (3 g, dry basis) was weighed into a RVA sample canister and distilled water was added to prepare a suspension with a total moisture of 12% for RVA testing.

### 2.4. Physicochemical Analysis of Waxy Rice Flour

The moisture content of samples was analyzed by using a Lab Oven at 105 °C for 2 h. The protein content was determined by the Kjeldahl method and the nitrogen (N) value was 5.95. Ash content was analyzed by muffle furnace and fat content of rice samples was measured following Soxhlet extraction. Amylose content of rice flour was determined using the iodine binding method by Leewatchararongjaroen et al. [16]. All measurements were performed in triplicate.

Damaged starch content was determined according to the American Association of Cereal Chemists 76-30A method (AACC 2000). The particle size distribution and average particle size of the flours were determined by a Mastersizer (Malvern Instrument, Malvern, UK), which was equipped with laser beam to detect the individual particles.

### 2.5. X-ray Diffraction

The X-ray diffraction pattern (XRD) of rice flour was conducted by an X-ray diffractometer (Bruker, D8 PHASER, Berlin, Germany) equipped with a copper tube operating at 40 kV and 200 mA, and producing Cu-Kα radiation of 0.154 nm wavelength. Diffractograms were obtained by scanning from 5° to 40° (2θ) at a rate of 4°/min and a step size of 0.03° at room temperature. The relative crystallinity (RC) was calculated according to the following equation [19]:RC = (Ac)/(Ac + Aa) × 100%,(1)
where Ac is the area of the crystalline peak, and Aa is the area of the amorphous peak. All measurements were performed in triplicate.

### 2.6. Determination of Structural Features of Isolated Starches

#### 2.6.1. Isolation of Rice Starch

The isolation of starch followed a method described by De et al. [20] with slight modifications. Rice flour (20 g, dry basis) was firstly mixed with 100 mL of 0.2% NaOH solution for 4 h at 37 °C, and then centrifuged at 1600× *g* for 10 min. The supernatant was discarded, and the upper yellow layer was scraped from the starch. A total of 100 mL 0.2% NaOH solution was added to the sediment and the above extraction operation was repeated. Then, the obtained sediment was washed five times using distilled water, followed by freeze-drying. Finally, the starch samples (IWR/JWR-number-S, the number stands for sample number) were stored in a glass bottle before further analysis.

#### 2.6.2. Molecular Weight and Radius of Rotation Analysis

The weight-average molecular weight (M_w_) and the z-average radius of rotation (R_z_) of the starch samples were measured by a HPSEC–MALLS–RI system according to Liu et al. [21]. The HPSEC–MALLS–RI system contained a pump (LC-20AB, Shimadzu Corporation, Kyoto, Japan), a refractive index detector (Waters 2414, Waters Corporation, Milford, MA, USA), and a multi-angle laser light-scattering detector (DAWN EOS, Wyatt Tech. Corp., Santa Barbara, CA, USA). The starch samples (20 mg) were dispersed in 10 mL of 50 mmol/L NaNO_3_-dimethyl sulfoxide (DMSO) solution. Subsequently, the suspensions were heated in boiling water for 1 h and kept at 50 ℃ with stirring for 24 h. The solution was then passed through a 0.22 μm membrane filter. The mobile phase was 50 mmol/L NaNO_3_-DMSO, which had been filtered through a 0.22 μm filter. The flow rate of mobile phase was 0.6 mL/min. Data obtained from MALLS and RI detectors were analyzed by Astra software (Version 5.3.4.20, Wyatt Tech. Corp., Santa Barbara, CA, USA).

#### 2.6.3. Fine Structure of Amylopectin

The average chain length and chain length distributions (CLDs) of amylopectin were determined by high performance anion exchange chromatography (DIONEX ICS-5000+SP-5), referring to the method of Han et al. [22] with some modifications. Starch powder (50 mg) was suspended in 5 mL of a 90% (*v*/*v*) DMSO solvent. The solution was heated in a boiling water bath for 1 h. Then, 2 mL of the solution was mixed with 12 mL of absolute ethanol. The starch molecules were recovered by centrifugation at 4000× *g* for 10 min, which was followed by resuspension in 5 mL preheated (90 °C) glycine-HCl buffer (pH 3.5). Two μL of isoamylase was added after it was cooled to 50 °C, and then incubated at 50 °C for 48 h [23]. The debranched starch samples were filtered through a 0.45 μm filter (Agilent Technologies, Santa Clara, CA, USA). All measurements were done in triplicate.

### 2.7. Removal of Lipid, and Lipid and Protein from Rice Flours

#### 2.7.1. Removal of Lipid

The lipids of waxy rice flour were extracted by petroleum ether [24]. The waxy rice flour (50 g) was mixed with 250 mL of petroleum ether, continuously stirring at 25 °C for 6 h. Filtration was conducted, and solid residue was obtained. The process was repeated twice and waxy rice flour without lipids (IWR/JWR-number-L, the number stands for sample number) was obtained. The pasting properties were tested as described in Section 2.3.

#### 2.7.2. Removal of Lipid and Protein

Waxy rice flour without lipid and protein was prepared sequentially. The proteins in the defatted flour were removed with alkaline protease [25]. Waxy rice flour of 50 g weight was mixed with 400 mL of alkaline protease solution (120 U/mL) in Na_2_CO_3_ buffer (0.02 M, pH 9.0). The suspension was hydrolyzed at 45 °C for 2 h with continuous stirring, and then it was centrifuged at 1600× *g* for 15 min. The solid residue was hydrolyzed again and the residues were washed with distilled water to neutral pH. Finally, waxy rice flour without lipid and protein (IWR/JWR-number-L-P, the number stands for sample number) was dried and stored in a glass bottle. The pasting properties were tested as described in Section 2.3.

### 2.8. Pasting Properties of the Isolated Rice Starch with Protein Addition

The proteins in IWR and JWR were extracted by 0.05 mol/L aqueous sodium hydroxide solution described by Yi et al. [26]. Protein was mixed with JWR-3-S at 10% concentration of starch weight, then the mixture was added to 1.3 mL volume of water to prepare 12% (*w*/*w*, dry basis) starch suspension. The pasting properties were tested as described in Section 2.3.

During the test process, JWR-3-S in the presence, or absence, of protein was collected at peak viscosity. The morphological characteristics of samples were examined using a VHX-1000C type Ultra Depth Microscope from KEYENCE Co., (Osaka, Japan). The particle size distribution was determined using a Laser Particle Size Analyzer (Mastersizer 3000; Malvern Instruments, Malvern, UK).

### 2.9. Extraction of Waxy Rice Protein

Flour samples were defatted before protein extraction to minimize lipid contamination. Albumin, globulin, glutelin and prolamin were extracted by deionized water, 5% NaCl solution, 0.02M NaOH solution and 70% alcohol, respectively [27]. Briefly, the defatted flour (100 g) was extracted by means of 400 mL distilled water for 4 h (albumin extract) and then centrifuged at 3000× *g* for 30 min. The residue was extracted with 400 mL of 5% NaCl for 4 h (globulin extract) and then centrifuged at 3000× *g* for 30 min. The residue was extracted with 300 mL of 0.02 M NaOH for 30 min (glutelin extract), and followed by 300 mL of 70% ethanol for 4 h (prolamin extract). Each extraction was conducted twice in order to remove all the protein of each fraction.

### 2.10. Pasting Properties of the Isolated Rice Starch with Globulin and Glutelin Addition

Globulin and glutelin were extracted from the waxy rice flour as described above. The extraction was dialyzed by a dialysis bag with cut-off molecular weight of 1000 Da for 48 h at room temperature. The dialyzed protein solutions were lyophilized. Protein contents of the lyophilized protein fractions were analyzed by the Kjeldahl method as in Section 2.4, and they were found to be higher than 95%. Each protein assay was conducted in triplicate. Meanwhile, the protein fractions were analyzed by SDS-PAGE. The results shown in Appendix A confirmed that the main protein in the protein fractions were globulin and glutelin, respectively.

Globulin and glutelin were mixed with JWR-3-S at 10% concentration of starch weight, respectively. Then the mixture was added to 1.3 mL volume of water to prepare 12% (*w*/*w*, dry basis) starch suspension. The pasting properties were tested as described in Section 2.3.

### 2.11. Statistical Analysis

The data were analyzed by analysis of variance (ANOVA) and presented as the average value ± standard deviation of triplicate determinations. The significant difference between the experimental data was analyzed by Duncan’s multiple comparison method (*p* < 0.05). Origin 2018 software was used for graphing and data analysis was performed using IBM SPSS 22.0 software.

## 3. Results and Discussion

### 3.1. Pasting Properties of Waxy Rice Flours

The pasting properties of starch samples are usually determined with a RVA [25] or a rotational rheometer [28,29]. In this study, the pasting properties of nine flours were tested using a rotational rheometer and the pasting profiles are shown in Figure 1. The pasting parameters, including peak viscosity (PV), trough viscosity (TV), final viscosity (FV) and pasting temperature (PT), are reported in Appendix A.

Waxy rice flours of the same variety have similar viscosity curves. The five varieties of IWR-F have similar FV, but differ in PV and TV. Among the IWR-F, IWR-1-F exhibited the highest value of PV, while IWR-2-F was lower, which may be related to the total starch content in the waxy rice flour. IWR-1-F had the highest total starch content among the IWR-F. The TV was affected by the M_w_ and R_z_ of starch [30]. Although IWR-2-S had a higher M_w_, its R_z_ was lower, and the degree of internal interaction during the starch gelatinization was reduced, so it had a lower TV. Among JWR-F, JWR-1-F and JWR-2-F have similar pasting viscosity curves, both higher than that of JWR-3-F and JWR-4-F. It was also found that the gelatinization temperatures of the samples decreased with the decrease of amylose content in both IWR-F and JWR-F. The ordered crystalline regions of amylose molecules require more energy to gelatinize, so starch with high amylose content has a higher gelatinization temperature [31].

In the two cultivars of waxy rice flour, it could be clearly seen that there were obvious differences between IWR-F and JWR-F in pasting parameters. The PV, TV, FV and PT of IWR-F were in the ranges of 1242 to 1371 cP, 833 to 899 cP, 1116 to 1174 cP and 70.31 to 72.39 °C, respectively, which were all significantly higher than those of JWR-F which ranged from 667 to 904 cP, 497 to 669 cP, 656 to 856 cP and 64.21 to 68.32 °C accordingly. These results were in a good agreement with previous investigations which reported that IWR flour has higher gelatinization viscosity than JWR flour [18]. The pasting profiles of the nine samples were also evaluated using a RVA, and the same data trends have been found in the experiment (shown in Appendix A). The pasting properties of waxy rice flour are very important parameters in evaluating the potential applications of respective rice flours in related and non-related food industries [32]. Therefore, it is necessary to study the parameters that cause the pasting properties to differ between indica and japonica waxy rice.

### 3.2. Physicochemical Properties of Waxy Rice Flour

The chemical compositions of nine flours are shown in Table 1. The contents of fat, ash and protein ranged from 0.24 to 0.69%, 0.26 to 0.44% and 7.27 to 10.50%, respectively. For all samples, total starch contents varied from 87.76 to 92.86%, with JWR-2-F recording the highest total starch content and JWR-3-F produced the lowest total starch content. These results were within the ranges previously reported by Li et al. [33]. In general, there was no significant difference in protein, fat, ash and starch contents between the two cultivars. The contents of amylose for both IWR-F and JWR-F were less than 5%, consistent with other research analyzed by Keeratipibul et al. [34], which found that there was little amylose in waxy rice flour. However, the amylose content of JWR-F samples were in the range of 3.18 to 3.68%, a little lower than the contents of IWR-F (4.08 to 4.58%). The difference in the pasting viscosity between IWR and JWR may be attributed to the difference in amylose content. It has been reported that low molecular mass amylose can leach out of the granules into the solution, and then form a viscous paste, which may contribute to an increase in the pasting viscosity [35].

Grinding is a unit operation to obtain waxy rice flour of fine particles. The damaged starch and particle size of waxy rice flour are two important factors affecting the physicochemical properties and the applications of waxy rice flour [36]. Asmeda et al. [32] found that the percentage of damaged starch granules had a negative relationship with pasting temperature, while the average particle size showed a strong negative relationship with gelatinization temperature. The starch damage and average particle size of waxy rice flour are reported in Table 1. The starch damage content was in the range of 8.26 to 12.48% and the average particle size was in the range of 76.65 to 164.8 μm. These results are in good agreement with a previous investigation that showed that starch damage content and range of particle size of dry grinding rice flour were 10.70% and 0.74–178.15 μm, respectively [32].

The XRD diffractograms of the two cultivars of waxy rice flours are shown in Figure 2. Five IWR-F and four JWR-F showed the same X-ray diffraction pattern. The waxy rice starch displays an A-type pattern with a doublet at 17 and 18° and two singlets at 15 and 23°of 2θ [37]. The relative crystallinities of five IWR-F and four JWR-F were in the ranges of 31.62 to 35.62% and 30.17 to 35.98%, respectively, which were at the same level as reported relative crystallinity of 34.6% for waxy rice starch [3]. There was no significant difference in relative crystallinity between the two cultivars. The peak at 2θ = 20° suggests that an amylose–lipid complex (V-type) is present in starch granules [25]. Herein, two cultivars of waxy rice flours exhibited a very weak diffraction peak at 20°, which was due to the lower amylose and lipid contents in waxy rice flours.

Correlation analysis between physical-chemical properties of waxy rice flour and pasting profiles was performed. Pearson correlation coefficients indicated that only amylose contents were correlated (*p* < 0.05) with pasting parameters (data not shown). This result was in agreement with amylose content being positively correlated to pasting temperature [2,9]. However, the difference of amylose content between the two cultivars was only about 1%. Therefore, the slight difference in amylose content may not be the main reason for the huge difference in pasting viscosity of the two cultivars of waxy rice.

### 3.3. Structural Information of Isolated Starches

As shown in Table 1, starch is the main component in waxy rice. Therefore, the basic properties (M_w_ and R_z_) and CLDs of waxy rice starch were investigated and the results are shown in Table 2. The M_w_ of waxy rice starch was from 2.75 to 4.57 × 10^7^ g/mol, and only a single peak was observed in the M_w_ spectrum (data not shown). These results were in good agreement with a previous investigation that showed that the M_w_ of native waxy rice starch was 2.12 × 10^7^ g/mol [38]. The R_z_ of nine starches was in the range of 121.6 to 135.1 nm, similar with a previous study that showed that the R_z_ of waxy rice power starch was 135.1 nm [39]. For M_w_ and R_z_, there was no obvious difference between the two kinds of starch. These results align with those of previous studies reported by Tappiban et al. [40] who analyzed other cereal starches.

The average chain length (DP) of nine starches were all about 16%, and no significant differences among the two waxy rice cultivars were observed. The CLDs were fractionated into four different classes as A (6 ≤ DP ≤ 12), B1 (12 < DP ≤ 24), B2 (24 < DP ≤ 36) and B3 chains (DP > 36). The contents of A, B1, B2, and B3 were in the ranges of 33 to 35%, 53 to 55%, 8 to 9% and 2%, respectively. In general, there were no significant differences between the two cultivars of starches in fine structure. Correlation analysis between basic properties of waxy rice starch and pasting profiles was performed. Pearson correlation coefficients indicated that all pasting parameters evaluated were not correlated (*p* < 0.05) with basic properties of waxy rice starch.

### 3.4. Effect of Lipid and Protein Removal on the Pasting Properties of Waxy Rice Flours

The pasting curves of waxy rice flours after removing the lipid were measured and results are displayed in Figure 3a,b. The pasting parameters, including PV, TV, FV and PT, are reported in Appendix A. The PVs of 5 IWR-L were in the range of 1429 to 1529 cP, which were all significantly higher than those of the 4 JWR-L which ranged from 917 to 1100 cP. The PV increase of flour after removing the lipid are shown in Figure 4a. The viscosity increase of IWR-L was in the range of 131 to 193 cP, and the viscosity increase of JWR-L was in the range of 84 to 280 cP, which is consistent with the results of a previous study that the PV of highland barley flour increased from 983 to 1103 cP after removing the lipid [25]. This result may be attributed mainly to the amylose–lipid complex, which is formed during heating and exerts an inhibitory effect on the swelling of starch.

After removing the lipid and protein, the pasting curves of waxy rice starches were measured and results are displayed in Figure 3c,d. The pasting parameters, including PV, TV, FV and PT, are reported in Appendix A. The PV increase of flour after removing lipid and protein are shown in Figure 4b. The PV, TV and FV of IWR-L-P increased to the ranges of 1543 to 1632 cP, 925 to 1060 cP and 1113 to 1271 cP, respectively. Compared with the viscosity curves of IWR-F, the PV values of IWR-L-P only increased 214–332 cP. However, the PV of four JWR-L-P increased to the range of 1375 to 1592 cP, and increased significantly as compared with JWR-F. For example, the PV of JWR-3 increased from 667 to 1464 cP, and JWR-4 increased from 765 to 1592 cP. Moreover, after removing lipid and protein, there were no significant differences between the two cultivars of starches. Similar results have been reported by Ding et al. [41]. who indicated that, after removing the protein from adlay seed (Coix lacryma-jobi L.), the PV increased from 1978 to 2714 cP. In this study, the results suggested that protein played an important role in pasting characteristics. The removal of protein can significantly increase the pasting viscosity of JWR, while the pasting viscosity of IWR increased slightly.

### 3.5. Effect of Protein Addition on the Pasting Properties of Isolated Starch

In order to prove the effect of protein on the pasting behavior of waxy rice flour, protein was extracted from IWR-F and JWR-F, respectively, and then was added to the JWR-3-S at 10% concentration of starch weight. The pasting profiles were tested and results are shown in Figure 5. The pasting temperature was unchanged after adding two kinds of protein. The pasting curves of JWR-3-S decreased obviously when 10% protein was added, which meant that both IWR-protein and JWR-protein had an inhibitory effect on the pasting viscosity of japonica waxy starch. Besides this, the results showed that, by adding 10% IWR-protein to the JWR-3-S, the PV, TV and FV decreased to 1224, 474 and 812 cP, respectively. However, when adding 10% JWR-protein to the JWR-3-S, the PV, TV and FV decreased to 1066, 297 and 562 cP, respectively, which were obviously lower than adding 10% IWR-protein group.

The morphological characteristics and particle size distribution of JWR-3-S in the presence or absence of protein were examined and the results are shown in Figure 6. As shown in the morphological image, the volume of the starch granules decreased in the presence of JWR-protein and IWR-protein, and the magnitude of the decreasewas greater than that of the former. The particle size of the starch granules decreased from 42.97 ± 3.67 to 36.20 ± 2.21 μm of IWR-protein group and 30.80 ± 0.36 μm of JWR-protein group. The outcome meant that protein can form a network structure through disulfide bonds to limit the swelling of waxy rice starch during pasting, so there was an obvious decrease in the pasting viscosity of starch. This is consistent with the previous measurement results that Chen et al. [12] found, wherein the morphology of starch granules became more compact when glutenin or gliadin was added, and confirmed the inhibitory effects of glutenin or gliadin on the gelatinization of starch.

As shown in Figure 1, the PV of JWR-3-F was 667 cP, which is significantly lower than 1224 cP of adding 10% JWR-protein group. This phenomenon indicated that endogenous protein in japonica waxy rice had a more obvious inhibitory effect on the pasting viscosity than exogenous protein. Previous studies suggested that starch granule-associated proteins are present both in the channels and on the surfaces of starch granules, named granule-channel proteins and granule-surface proteins [42].

### 3.6. Analysis of Protein Composition in Waxy Rice Flour

The above results suggest that protein is the main cause of the pasting properties of IWR-F being significantly different from those of JWR-F. According to previous results, the protein in rice flour can be roughly divided into glutelin, globulin, albumin and prolamin according to molecular weight and solubility [27,43]. In order to investigate the protein composition, the contents of glutelin, globulin, albumin and prolamin in the IWR-F and JWR-F were analyzed, respectively, and the results are shown in Figure 7. As seen in Figure 7, the total contents of glutelin, globulin, albumin and prolamin were higher than 95%, and the contents of other proteins were less than 5%. A similar result has been reported that the total content of the four proteins in rice flour were higher than 97% of the protein content [27]. The content of glutelin in the sample was the highest, which was consistent with other results [44]. As seen in Figure 7, there were obvious differences in the content of glutelin and globulin between IWR-F and JWR-F. The content of glutelin in the IWR-F was about 64%, significantly lower than 73% of JWR-F, while the content of globulin in the IWR-F was about 18%, significantly higher than 11% of JWR-F.

### 3.7. Effect of Glutelin and Globulin Addition on the Pasting Properties of Isolated Starch

In order to prove the effect of glutelin and globulin on the pasting behavior of waxy rice flour, glutelin and globulin were extracted from waxy rice flour, and then added to the JWR-3-S at 10% concentration of isolated starch weight, respectively. The pasting profiles were tested and results are shown in Figure 8. The pasting curves of JWR-3-S decreased obviously when glutelin and globulin were added, which meant that both glutelin and globulin had an inhibitory effect on the pasting viscosity of isolated starch. Besides this, the results showed that by adding 10% globulin to the isolated starch, the PV, TV and FV decreased to 1290, 646 and 904 cP, respectively. However, when adding 10% glutelin to the starch, the PV, TV and FV decreased to 992, 503 and 728 cP, respectively, which were obviously lower than adding 10% globulin group, 10% IWR-protein group and 10% JWR-protein group (In Figure 5). The effects of adding glutelin and globulin to rice starch on its pasting properties have been studied by Baxter et al. [11], and the results of their study found that adding glutelin and globulin decreased the pasting viscosities of starch, but the magnitude of glutelin was greater than globulin. For example, the addition of glutelin and globulin decreased the peak viscosity by about 35 RVU and 16 RVU, respectively.

In order to quantify the effects of glutelin and globulin on pasting properties, the relationships between PV, TV and FV of waxy rice flour and the content of globulin and glutelin were investigated and the results are shown in Appendix A. The PV, TV and FV of waxy rice flour showed a positive relationship with the content of globulin, the range of R values was from 0.94 to 0.98. However, the PV, TV and FV of waxy rice flour showed a negative relationship with content of the glutelin; the range of R values was from 0.94 to 0.98. Therefore, the high content of glutelin and low content of globulin in JWR-F may be the main causes for the pasting parameters of JWR-F to be lower than those of IWR-F.

## 4. Conclusions

The objective of this study was to analyze the difference between the pasting properties of five IWR-F and four JWR-F. The results suggested that protein played an important role in the pasting characteristics. The addition of both JWR-protein and IWR-protein decreased the viscosity, volume and particle size of starch granules, but the magnitude of the decreases of JWR-protein were greater than for IWR-protein. Differences in protein composition between five IWR-F and four JWR-F come into focus. The content of glutelin in the IWR-F was significantly lower than JWR-F, while the content of globulin in the IWR-F was significantly higher than JWR-F. The addition of both glutelin and globulin decreased the pasting viscosities, but the magnitude of the decreases due to glutelin were substantially greater than for globulin. Therefore, a higher content of glutelin and a lower content of globulin in JWR-F may be the main reason causing the pasting parameters of JWR-F to be lower than those of IWR-F.

## Figures and Tables

**Figure 1 foods-11-01132-f001:**
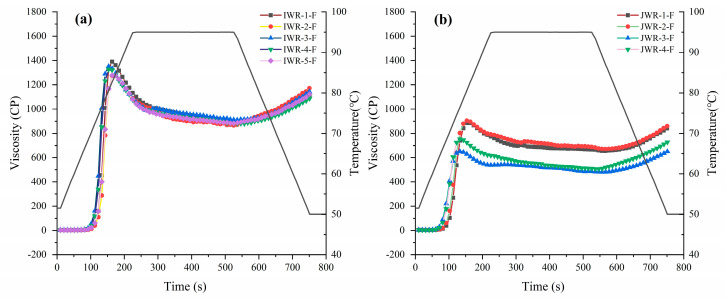
The pasting characteristics of five IWR flours (**a**) and four JWR flours (**b**). “IWR” stands for indica waxy rice; “JWR” stands for japonica waxy rice. “F” stands for flour.

**Figure 2 foods-11-01132-f002:**
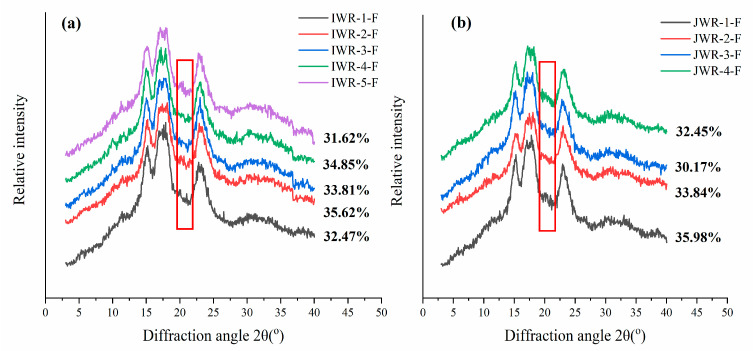
X-ray diffractograms of IWR-F (**a**) and JWR-F (**b**). “IWR” stands for indica waxy rice; “JWR” stands for japonica waxy rice. “F” stands for flour.

**Figure 3 foods-11-01132-f003:**
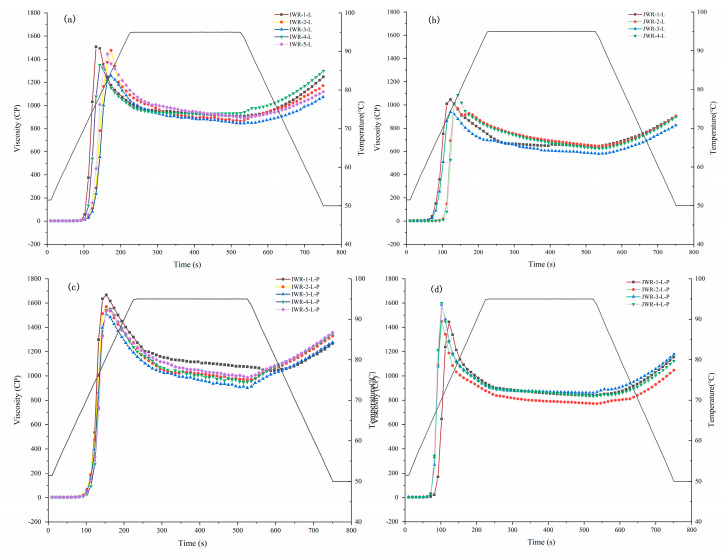
The pasting characteristics of five IWR flours (**a**,**c**) and four JWR flours (**b**,**d**) after removing lipid (**a**,**b**), lipid and protein (**c**,**d**). “IWR” stands for indica waxy rice; “JWR” stands for japonica waxy rice.

**Figure 4 foods-11-01132-f004:**
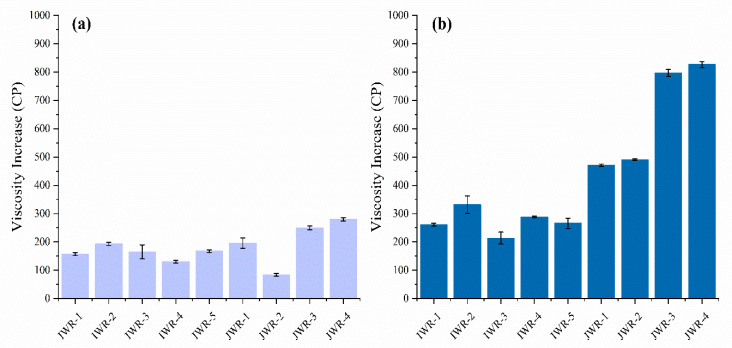
The peak viscosity increase of flour after removing lipid (**a**) and after removing lipid and protein (**b**). “IWR” stands for indica waxy rice; “JWR” stands for japonica waxy rice.

**Figure 5 foods-11-01132-f005:**
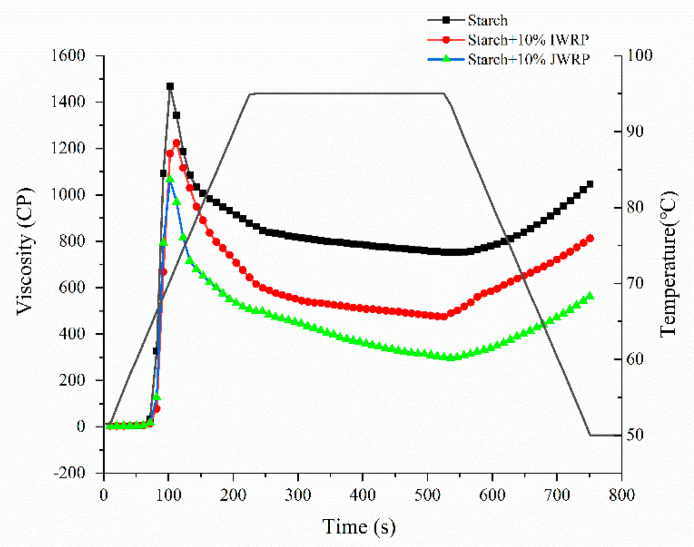
Pasting characteristics of waxy rice starch in the presence and absence of 10% protein extracted from indica and japonica waxy rice flour. The pasting curves were tested at 12% (*w*/*w*, dry basis) of starch suspension. “IWRP” stands for protein extracted from indica waxy rice flour.

**Figure 6 foods-11-01132-f006:**
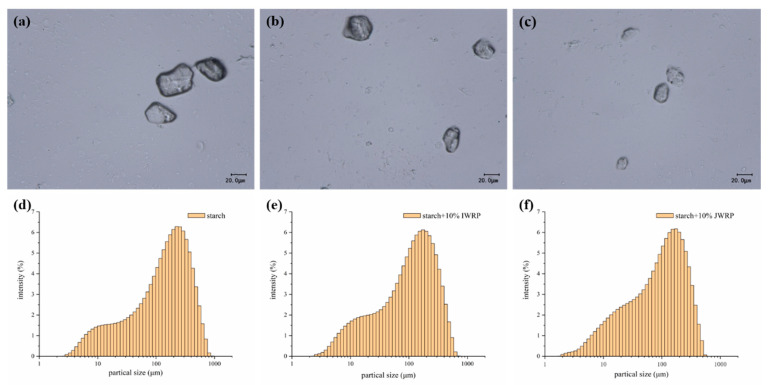
The morphological characteristics (**a**–**c**) and particle size distribution (**d**–**f**) of starch (**a**,**d**), starch + 10% IWRP (**b**,**e**) and starch + 10% JWRP (**c**,**f**). “IWRP” stands for protein extracted from indica waxy rice; “JWRP” stands for protein extracted from japonica waxy rice.

**Figure 7 foods-11-01132-f007:**
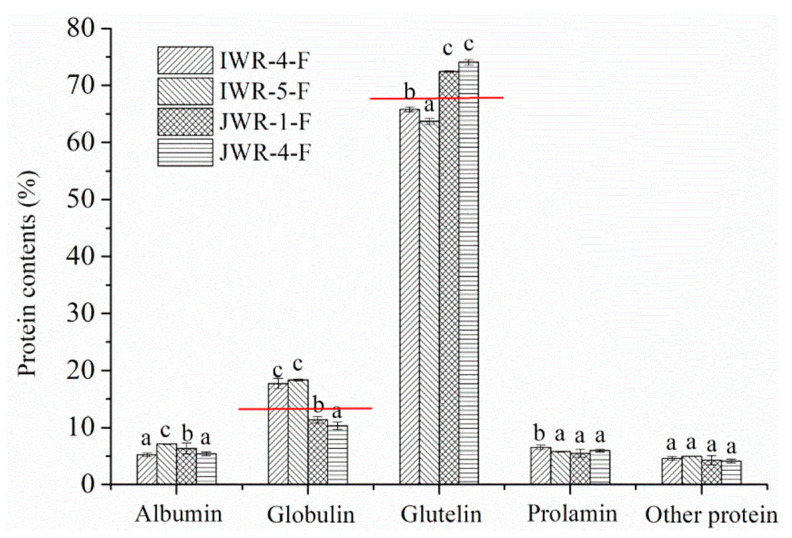
The contents of glutelin, globulin, albumin and prolamin in the IWR-F and JWR-F. “IWR” stands for indica waxy rice; “JWR” stands for japonica waxy rice. “F” stands for flour. The different letters for each index represent significant differences (*p* < 0.05).

**Figure 8 foods-11-01132-f008:**
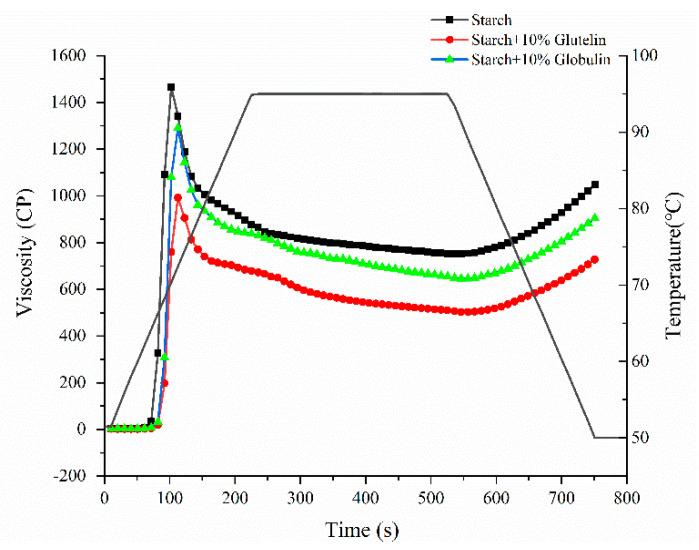
The effects of glutelin and globulin on the pasting properties of isolated waxy rice starch.

**Table 1 foods-11-01132-t001:** Physicochemical properties of waxy rice flour (dry basis).

Sample	Protein (%)	Fat (%)	Total Starch (%)	AMYLOSE (%)	Ash (%)	Starch Damage (%)	Average Particle Size (μm)
IWR-1-F	8.83 ± 0.17 ^d^	0.53 ± 0.01 ^e^	89.77 ± 0.00 ^b^	4.58 ± 0.09 ^d^	0.29 ± 0.01 ^b^	9.40 ± 0.11 ^bc^	164.33 ± 0.88 ^e^
IWR-2-F	7.27 ± 0.14 ^a^	0.58 ± 0.02 ^f^	92.86 ± 0.14 ^d^	4.44 ± 0.05 ^d^	0.26 ± 0.00 ^a^	8.26 ± 0.21 ^a^	163.33 ± 1.89 ^e^
IWR-3-F	8.30 ± 0.00 ^c^	0.56 ± 0.02 ^ef^	91.05 ± 1.40 ^c^	4.13 ± 0.09 ^c^	0.35 ± 0.01 ^c^	11.42 ± 0.07 ^f^	76.65 ± 1.96 ^a^
IWR-4-F	7.46 ± 0.26 ^ab^	0.44 ± 0.015 ^d^	91.58 ± 0.62 ^c^	4.13 ± 0.09 ^c^	0.35 ± 0.01 ^c^	9.61 ± 0.10 ^cd^	140.17 ± 2.14 ^d^
IWR-5-F	7.90 ± 0.10 ^bc^	0.41 ± 0.02 ^d^	90.83 ± 1.23 ^bc^	4.08 ± 0.05 ^c^	0.36 ± 0.01 ^c^	10.43 ± 0.10 ^e^	164.83 ± 7.41 ^e^
JWR-1-F	7.67 ± 0.11 ^ab^	0.28 ± 0.01 ^b^	89.90 ± 0.60 ^b^	3.68 ± 0.00 ^b^	0.34 ± 0.01 ^c^	12.03 ± 0.14 ^gh^	98.47 ± 1.94 ^b^
JWR-2-F	8.13 ± 0.10 ^c^	0.24 ± 0.005 ^a^	91.69 ± 0.57 ^c^	3.54 ± 0.05 ^b^	0.37 ± 0.01 ^c^	12.48 ± 0.31 ^hi^	113.28 ± 5.90 ^c^
JWR-3-F	9.41 ± 0.18 ^e^	0.69 ± 0.00 ^g^	87.76 ± 2.14 ^a^	3.27 ± 0.05 ^a^	0.44 ± 0.00 ^d^	11.62 ± 0.14 ^fg^	86.73 ± 0.53 ^a^
JWR-4-F	10.50 ± 0.02 ^f^	0.36 ± 0.00 ^c^	89.95 ± 1.70 ^b^	3.18 ± 0.05 ^a^	0.42 ± 0.00 ^d^	11.99 ± 0.23 ^gh^	81.87 ± 1.76 ^a^

“IWR” stands for indica waxy rice; “JWR” stands for japonica waxy rice; “F” stands for flour; Values are shown by Mean ± SD and values, different letters within a column indicate significant differences between mean values (*p* < 0.05).

**Table 2 foods-11-01132-t002:** Structural features of isolated waxy rice starches.

Sample	M_w_/ × 10 ^7^ g/mol	R_z_/nm	Average DP/%	A Chain/%	B_1_ Chain/%	B_2_ Chain/%	B_3_ Chain/%
IWR-1-S	3.91 ± 0.00 ^f^	133.5 ± 0.1 ^f^	16.46 ± 0.00 ^e^	34.18 ± 0.00 ^c^	54.17 ± 0.00 ^g^	9.23 ± 0.01 ^f^	2.42 ± 0.00 ^b^
IWR-2-S	4.57 ± 0.00 ^h^	128.8 ± 0.2 ^e^	16.46 ± 0.00 ^e^	34.02 ± 0.01 ^b^	54.09 ± 0.01 ^e^	9.77 ± 0.00 ^h^	2.12 ± 0.01 ^a^
IWR-3-S	3.06 ± 0.01 ^c^	126.8 ± 0.2 ^d^	16.47 ± 0.00 ^e^	34.44 ± 0.01 ^d^	53.91 ± 0.01 ^d^	8.94 ± 0.00 ^d^	2.71 ± 0.00 ^h^
IWR-4-S	3.94 ± 0.01 ^g^	135.1 ± 0.1 ^g^	16.48 ± 0.01 ^f^	34.52 ± 0.01 ^e^	53.51 ± 0.01 ^a^	9.50 ± 0.01 ^g^	2.47 ± 0.01 ^d^
IWR-5-S	3.28 ± 0.03 ^e^	126.6 ± 0.1 ^d^	16.61 ± 0.01 ^g^	33.25 ± 0.01 ^a^	55.10 ± 0.01 ^h^	9.03 ± 0.01 ^e^	2.61 ± 0.01 ^f^
JWR-1-S	2.75 ± 0.01 ^a^	121.6 ± 0.2 ^a^	16.32 ± 0.01 ^d^	34.80 ± 0.00 ^f^	54.11 ± 0.00 ^f^	8.43 ± 0.01 ^c^	2.66 ± 0.01 ^g^
JWR-2-S	3.20 ± 0.02 ^d^	124.2 ± 0.2 ^b^	16.28 ± 0.00 ^c^	34.94 ± 0.01 ^g^	54.12 ± 0.01 ^f^	8.35 ± 0.00 ^b^	2.59 ± 0.01 ^e^
JWR-3-S	3.19 ± 0.09 ^d^	125.3 ± 0.3 ^c^	16.19 ± 0.01 ^b^	35.52 ± 0.01 ^h^	53.71 ± 0.01 ^c^	8.35 ± 0.00 ^b^	2.42 ± 0.01 ^b^
JWR-4-S	3.04 ± 0.01 ^b^	124.2 ± 0.2 ^b^	16.17 ± 0.00 ^a^	35.59 ± 0.01 ^i^	53.65 ± 0.01 ^b^	8.32 ± 0.01 ^a^	2.43 ± 0.00 ^c^

“IWR” stands for indica waxy rice; “JWR” stands for japonica waxy rice. “S” stands for isolated starch; Values are showed by Mean ± SD and values; Different letters within a column indicate significant differences between mean values (n = 3) (*p* < 0.05).

## Data Availability

Data are contained within the article and Appendix A.

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
