# Peer review of "Study on the Pasting Properties of Indica and Japonica Waxy Rice"

_foods, 2022, doi:10.3390/foods11081132_

Round 1

Reviewer 1 Report

The manuscript has sound research methodology to explain the potential factors responsible for  pasting properties of different varieties of waxy rice. However, the manuscript lacks sufficient depth in the introduction to justify rationale and objectives. Methodology on the whole is detailed but can be further improved. Most part the results are presented but not well discussed except comparing with similar findings of other work. I believe results and discussion section can be further improved. 

Please find my comments directly provided on the attached PDF.

Reviewer 2 Report

The manuscript is well written and authors have done good research work and efforts in developing the manuscript. Their are minor changes that should be considered by the authors as stated in the attached file.Some important changes are suggested below:

Title of the manuscript could use revision.

Introduction: Simplify to include active sentences at few instances denoted, authors are encouraged to review the entire manuscript to address this. A thesis statement is must for an introduction and a well performed work such as this.

Material and methods: well written and explicit

Result and discussion: minor changes on unit representation suggested.

Conclusion: this section could use some revision, suggested in attachment

Reference: format according to journal specifications and formats

Reviewer 3 Report

The study of Fang et al. is interesting, the experiment performed is rather complex and appears well executed. The manuscript need revision before final decision:

Lines 21-23: The following statement needs revision: “Meanwhile, the addition …IWR-protein.”

Line 23: I suggest replacement with “The IWR protein contained about …”

Lines 53-54: Rephrasing is needed.

In many places the use of the verb “differ” is not appropriate. I suggest replacing it with “impact” or “affect”

Line 62: It is not clear how the results could impact the waxy rice-related manufacturers.

Lines 67 and 69: The IWR and JWR were previously defined.

All abbreviations must be defined when first used in the manuscript body. Take for instance JWR-3-S which was not clearly defined. Please carefully check the entire manuscript.

The measurements of the pasting behavior of the samples by means of the rapid visco analyzer were not mentioned in the Materials and methods section.

Figure 4: It should be clearly mentioned that the increase refers to the peak viscosity.

Line 346: It is not clear what the authors meant by “addition of two kinds of protein did not lead to the reduction of water.” Please revise or elaborate.

I suggest moving the Figure 9 form the manuscript to the Supplementary material. It is enough to provide the R2 values in the manuscript body.

English correction needed, preferentially by a professional.

Pay attention to punctuation. In several places, comma must be used instead of semicolon.

Round 2

Reviewer 1 Report

The authors have addressed all the comments and suggestions. 

Reviewer 3 Report

The manuscript was improved and can be accepted for publication.